# Physical Activity Recommendations in the Context of New Calls for Change in Physical Education

**DOI:** 10.3390/ijerph18031177

**Published:** 2021-01-28

**Authors:** Karel Frömel, Jana Vašíčková, Krzysztof Skalik, Zbyněk Svozil, Dorota Groffik, Josef Mitáš

**Affiliations:** 1Faculty of Physical Culture, Palacký University Olomouc, Tr. Miru 115, 77111 Olomouc, Czech Republic; karel.fromel@upol.cz (K.F.); jana.vasickova@upol.cz (J.V.); zbynek.svozil@upol.cz (Z.S.); 2Institute of Sport Sciences, The Jerzy Kukuczka Academy of Physical Education, Mikolowska 72A, 40-065 Katowice, Poland; k.skalik@awf.katowice.pl (K.S.); d.groffik@awf.katowice.pl (D.G.)

**Keywords:** steps, recommendation, wearables, online physical education, pandemic

## Abstract

The current social, health, and educational changes in society require an adequate response in school-based physical activity (PA), including physical education (PE) lessons. The objective of this study was to identify the real average step counts of Czech and Polish adolescents during PE lessons, and propose recommendations for improving PE programs. This research was carried out in 143 Czech and 99 Polish schools. In the research, a total of 4911 adolescents aged 12–18 years were analyzed as part of teaching practice and 1827 in the context of habitual school practice. Steps were monitored using pedometers. The average step count per PE lesson was 2390 in Czech and Polish boys, while girls achieved 1851 steps. In both countries, boys were subject to greater physical strain in PE lessons compared to girls, both in teaching practice (*F*(4088,3) = 154.49, *p* < 0.001, *η_p_*^2^ = 0.102) and school practice (*F*(1552,3) = 70.66, *p* < 0.001, *η_p_*^2^ = 0.103). Therefore, the priority in PE lessons is to increase the amount of PA for girls, achieve the objectives of PE during PA, and use wearables to improve awareness of PA and improve physical literacy, as well as to support hybrid and online PE as a complement to traditional PE.

## 1. Introduction

The acceleration of social, health, and educational changes requires adequate modifications to school physical education (PE). The response to the expansion of distance learning and the associated limitation of social contact, the increase in sedentary behavior, the increased mental strain, and the limited number of PE lessons and other activities that do not promote health should reflect the status and trends in the pre-pandemic period.

There has been a long-term downward trend in physical activity (PA) of adolescents [1,2], as well as an increase in sedentary behavior [3,4,5], overweight and obesity [6,7], insufficient cardiorespiratory fitness [8], and negative trends in mental health [9], including other undesirable effects on adolescents’ health.

In the Central European region, a decrease in PA [10], insufficient vigorous PA [11], and insufficient compensation for educational strain by PA [12,13] have been noted. The high cognitive and mental strain of adolescents in secondary schools is insufficiently compensated by PA during recess periods, after school, or at weekends [14,15]. An alarming finding is the occurrence of depression symptoms in 25% of Czech and Polish adolescents [16]. Moreover, the previously confirmed benefits of organized PA [17] are endangered in the time of bans on organized PA, including economic reasons, so a return to the original state will be difficult.

In most countries, PE struggles to improve its academic position among the other subjects in the curriculum. In many schools, education is considered in a narrow sense and is overly focused on knowledge-based academic achievement [18,19]. However, there is no doubt about the benefit of PE in supporting adolescents’ health [19,20,21]. Nevertheless, the evidence of the benefit of PE in promoting adolescents’ health and improving their educational outcomes is insufficiently reflected in educational reforms.

It has been confirmed that PE lessons contribute to an increase in moderate-to-vigorous physical activity (MVPA) in children [22] as well as in adolescents [20]. It has also been confirmed that a higher number of PE lessons per week is associated with higher weekly MVPA in adolescents [23]. In school programs, PE lessons increase daily MVPA by 12.8 min compared to days without PE lessons [24]. Similarly, in intervention programs in elementary and secondary schools, PE lessons contribute to a 24 p.p. increase in MVPA per day as opposed to days without PE lessons [25]. In terms of school PA, boys with PE lessons in the school program reported 749 ± 418 steps/school hour (without PE lessons = 371 ± 287 steps/school hour), but girls reported only 627 ± 354 steps/school hour (without PE lessons = 370 ± 273 steps/school hour) [14]. On days with PE lessons, boys accumulated 11,404 steps/day (girls 8301 steps/day), while on days without PE lessons, boys accumulated 9175 steps/day (girls 7238 steps/day) [26]. In secondary schools, during PE lessons, adolescents spend, on average, 35.9% (28.3–43.6%) of time in MVPA [20] and achieve 54% of their time use engaged in MVPA [27]. Similarly, according to telemetry measurement, 11–14 year-old adolescents spent 34.3 ± 21.8% of their PE lesson time engaged in MVPA [28]. In Central European countries, it has also been confirmed that PE lessons cannot be replaced by extending school recess time [14].

The recommendation that adolescents should spend 50% of PE class time in MVPA [29] is not achieved by most adolescents [30]. In this context, Scruggs [31] added that PE lessons should include 61.3 steps/min to achieve 30% of PE time engaged in MVPA and 83.9 steps/min to achieve 50% of PE time engaged in MVPA. In the Central European region, at least 20 min of MVPA is recommended for most types of PE lessons [11]. However, at least 20 min of MVPA (≥3 MET) in PE lessons was achieved by only 25.0% of Czech and Polish boys and 18.0% of Czech and Polish girls [11]. Based on practical experience and previous long-term research results, for most types of PE lessons, it is recommended to achieve submaximal or maximal heart rate at least once per PE lesson [14].

An alarming fact is that in the Czech Republic, there has been a statistically significant decrease in the evaluation of PE lessons by adolescents, particularly by those with lower sports and physical achievement [32]. Another serious fact is that the decrease in valuation is intensified by the deterioration of the “learner role” (being able to show off, engage, etc.) in PE lessons [32]. In the context of adequately responding to educational changes during the pandemic, it is also necessary to consider that in the Central European and Eastern European regions, PE teachers have insufficient experience with hybrid learning in PE and other forms of online PE. The lack of readiness for using online PE is obvious, especially in comparison to online PE in other countries such as South Korea or with analyses of the positives and negatives of online PE in the U.S. [33,34].

The pandemic has brought new challenges and requirements for lifestyle changes, including for adolescents. Therefore, it is imperative to quickly respond to the changes in secondary education. There is a serious risk that the differences in the achievement of cognitive and health objectives within the educational process will increase, just as the position of PE as a curricular subject. Thus, the critical considerations for simplifying the relationships between PE and health may deepen even more [35]. The same can be expected for the disputes between PE theory and PE practice. PE cannot effectively address the negative effects of the current pandemic on adolescents’ healthy lifestyle without being supported by school administration, parents, leisure institutions, and sports clubs. Despite this fact, its role is irreplaceable.

However, do we have sufficient knowledge about the current state and trends in PE to resolve the emerging socio-educational crises and respond to the new challenges adequately and effectively within new challenges and new tasks in school PE? Do we know the real level of PA in PE lessons expressed simply by step counts in the context of educational programs?

The objective of this study was to identify the real average step counts of Czech and Polish adolescents during PE lessons in natural educational environments and propose recommendations for improving PE programs.

## 2. Materials and Methods

### 2.1. Study Design and Participants

This cross-sectional research was carried out between 2015 and 2017 in 143 schools in the Czech Republic and 99 schools in Silesia in Poland. In total, the research involved 6738 adolescents aged 12–18 years (Table 1). A total of 12.2% of boys and 5.6% of girls were overweight or obese (14.1% of boys and 22.7% of girls were underweight).

This research was carried out in the framework of teaching practice and weekly PA monitoring in school practice:

(a) Monitoring of step counts in PE lessons as part of teaching practice (referred to as teaching practice):

This part of the research involved 107 schools in the Czech Republic and 58 schools in Silesia in Poland. The sample included various types of secondary schools (general, vocational, and apprentice) from different urban/rural locations, where teaching practices were carried out. In total, the research involved 4911 Czech and Polish adolescents. In both countries, all PE lessons were led by student teachers (fifth year of Masters degree program). The course of teaching practice was similar in both countries. The major difference lies in the PE curriculum where in Poland the education system includes three PE lessons per week, while the Czech Republic has only two PE lessons per week.

(b) Monitoring of step counts in PE lessons as part of weekly PA monitoring (referred to as school practice):

A total of 36 schools in the Czech Republic and 41 schools in Poland consented to the research-based monitoring of weekly PA. The main reason for the selection of schools and their consent to the research was long-term cooperation between the schools and the university departments. This part of the research study involved 1827 Czech and Polish adolescents. As part of the weekly PA monitoring, including PE, PE lessons were led by experienced PE teachers (qualified in PE and with more than five years of experience). In this part of the research, all participants, including their parents, provided informed consent for the research.

In total, 92% of PE lessons were carried out as single-sex PE, the remaining as coeducational or hybrid, combining both approaches (single-sex PE in parallel groups). Most student teachers and PE teachers in the Czech Republic were specialized in two disciplines (PE and a different subject), while in Poland, most PE teachers were specialized in just one discipline (only PE). The educational systems of both countries were similar. However, in the Czech Republic, there were usually only two PE lessons per week on average, while in Poland, there were four. In terms of teacher training, there were no significant conceptual differences, but Poland had a broader curriculum of theoretical and practical teaching subjects.

In both countries, the research was performed by the same research teams. The main requirement was to preserve the natural school environment and not to interfere with the school program or educational process. Another requirement was to avoid a narrow focus on step count monitoring in PE lessons, which typically does not reflect the usual level of PA in PE lessons. Secret research methods in PE lessons are ethically unacceptable. Therefore, the research method used in PE lessons was “addressing a secondary research problem.” In their teaching practices, student teachers led PE lessons in accordance with the school program and followed the usual educational objectives. In terms of PE objectives, the following areas were monitored on a regular basis: Cognitive, emotional, health, social, relationships, and creative areas and the comprehensive area of “pupil roles” in PE lessons (in compliance with the PE lesson questionnaire by Frömel et al. [32]). Similarly, in school practice, the pedometer-based monitoring of weekly PA included PE lessons. PA monitoring was not primarily focused on step counts in PE lessons, but on overall weekly PA. The recording of step counts in PE lessons represented one of the segments of the school day.

### 2.2. Procedures

During their teaching practice, student teachers monitored PA in a single PE lesson chosen by the supervising PE teacher according to actual educational organization (especially considering the length of recess periods before and after the PE lesson and the status of sports establishments). PE lessons were led independently by student teachers. Before the start of the PE lessons, the participants attached a pedometer to their waist and checked to ensure it showed a zero-step count in the pedometer. During the PE lessons, no checking of step counts or providing motivation to increase the number of steps was performed. In the final part of the PE lessons, during relaxation in the lying position, each participant recorded the step count they achieved in the PE lesson, as well as the pedometer number (due to data anonymity).

The introduction to weekly PA monitoring was held with the participants in a computer room one day prior to the commencement of PA monitoring. The participants were informed about the purpose of the research and the method of PA monitoring by using pedometers. They learned to manipulate the pedometers and to record the time and step counts for each segment of the day using the record sheet. The records also included the time and step counts at the beginning and at the end of the PE lessons. The participants registered on the International Database for Research and Educational Support web application (Indares, www.indares.com). More detailed information concerning the time and step counts for each segment of the day, including PE lessons, were entered into the web application from the record sheets. The PE lessons were led by PE teachers who were informed about the weekly PA monitoring. The objectives and content of the PE lessons were not affected by the research teams. The participants wore the pedometers throughout the whole week and removed it only for swimming, showering, and sleeping. The study included those respondents who participated in PE lessons during the monitoring week. The first PE lesson that met the inclusion criteria was counted. To compare teaching practice within countries, only step counts in PE lessons were used from the weekly PA monitoring.

### 2.3. Measurements

To determine the amount of PA in PE lessons, Yamax Digiwalker SW-700 pedometers were used (Yamax Corporation, Tokyo, Japan), which are also suitable for measuring shorter forms of PA such as PE [36]. The data concerning the number of steps (but also jumps, hops, and changes in body position) obtained from the pedometers were not adjusted in any way. Extremely low (<500 steps) and high (>6000 steps) step counts per PE lesson were eliminated. In total, 232 participants in teaching practice and 194 in school practice were eliminated. The pedometers were calibrated before each research block with a tolerance of 5%. In both countries, PE lessons lasted 45 min. Therefore, all data presented are based on PE/45 min.

### 2.4. Data Analysis

The data were analyzed in SPSS version 25 (IBM SPSS, Inc., Armonk, NY, USA) and Statistica version 13 (StatSoft, Prague, Czech Republic). Descriptive statistics were used to characterize the samples, difference tests to identify the differences in average step counts in PE lessons, crossing tables with Pearson’s χ^2^ to identify group differences in the achievement of the recommendations for PE lessons, one-way ANOVAs and post-hoc Scheffe tests to analyze group differences in step counts per PE lesson, and *η_p_*^2^ and r effect size coefficients. A logically significant difference was 500 steps in a 45 min PE lessons. The statistical significance was set at *p* < 0.05.

### 2.5. Ethical Statements

This study was approved by the Ethical Committee of the Faculty of Physical Culture, Palacký University Olomouc under No. 37/2013. The school administrations, parents, and participants confirmed their agreement to participate in the research by providing written consent. They were informed about the security and anonymity of the data obtained through the Indares web application, about the method of data processing, and about the further use of their data. The schools involved in the project and the participants were informed of the average group results.

## 3. Results

### 3.1. Average Step Count in PE Lessons in Teaching Practice and School Practice

The average step counts per PE lesson in teaching practice confirmed the already known facts about the higher amount of PA in PE lessons among Czech (*p* < 0.001) as well as Polish (*p* < 0.001) boys as opposed to girls (*F*(4907,3) = 184.60, *p* < 0.001, *η_p_^2^* = 0.101) (Figure 1). Similarly, in school practice, Czech (*p* < 0.001) and Polish (*p* < 0.001) boys reported, on average, more steps during PE lessons than girls (*F*(1823,3) = 70.86, *p* < 0.001, *η_p_^2^* = 0.104).

In comparison with their Polish counterparts, Czech boys reported more steps per PE lesson in teaching practice (*p* < 0.001) as well as in school practice (*p* = 0.007). As far as girls were concerned, no statistically significant differences were observed between girls from the two countries in teaching practice or school practice.

It is interesting to note that in terms of step counts in the PE lessons led by student teachers and PE teachers (Figure 1), no statistically significant differences were observed in the Czech (*p* = 0.897) or Polish (*p* = 0.948) boys. In the PE lessons led by student teachers, both Czech (*p* < 0.001) and Polish (*p* = 0.005) girls reported, on average, more steps per PE lesson than Czech and Polish girls in the PE lessons led by PE teachers.

### 3.2. Summary of Step Count Per PE Lesson among Boys and Girls

In all monitored PE lessons, boys reported (on average, 2390 ± 948 steps) statistically significantly more steps than girls (on average, 1851 ± 815 steps) (*F* = 563.48, *p* < 0.001, *η_p_^2^* = 0.087). Czech adolescents (on average, 2122 ± 1030 steps) reported more steps in PE lessons than Polish adolescents (on average, 2000 ± 723 steps) (*F* = 1.21, *p* = 0.271, *η_p_^2^* < 0.001), but the differences were not considered logically significant.

The recommendation of 2000 steps per PE lesson was achieved in teaching practice and school practice in 56.4–68.8% of the Czech and Polish boys, but only in 28.4–41.9% of girls (Figure 2). In total, the recommendation was achieved by 63.4% of the boys and 36.6% of the girls (χ^2^ = 465.96, *p* < 0.001, *r* = 0.263). The achievement of the 60 steps/min threshold was very low in both teaching practice and school practice. The recommendation was achieved in teaching practice and school practice in 16.4–43.4% of the Czech and Polish boys, but only in 9.4–16.6% of girls. In total, the recommendation of 60 steps/min was achieved by 31.7% of the boys and 13.2% of the girls (χ^2^ = 340.75, *p* < 0.001, *r* = 0.225). The highest achievement was observed in school practice among Czech boys (43.4%), while the lowest was observed in teaching practice among Polish girls (9.4%).

As far as younger and older adolescents in both countries are concerned, significant differences were observed only in Czech girls (*p* < 0.001) (Table 2). Compared with younger and older Czech girls, both younger (*p* = 0.021) and older (*p* < 0.001) Polish girls were more physically active. Younger Czech boys reported more steps per PE lesson than younger Polish boys (*p* < 0.001), but there were no statistically significant differences between older boys (*p* = 0.087). Similarly, no statistically significant differences were observed between younger Czech and Polish girls (*p* = 0.098) or older girls (*p* = 0.055).

In both countries, the differences between overweight and normal weight boys were not statistically significant; the same applied to girls (Table 3). The lowest physical activity in PE lessons was observed in Czech overweight girls, who, on average, reported only 1774 ± 959 steps, while the highest physical activity was observed in Czech boys (2603 ± 1086 steps).

## 4. Discussion

### 4.1. Step Counts of Boys and Girls in PE Lessons

The most serious finding of the study was the fact that the average step count per PE lesson in Czech and Polish boys was 2390 ± 948 steps and in girls was 1851 ± 815 steps, with a high degree of probability characterizing the amount of PA in habitual PE lessons in the Czech Republic and Poland. The insignificant differences in step counts per PE lesson among participants led by student teachers and PE teachers suggest that the lesser experience of the student teachers did not have a significant effect on the number of steps. As expected, the step counts were lower than those presented by partial research studies focused on dance PE lessons (2510–3047 steps) [37], team game lessons (59.8 ± 13.7 steps/min; 2691 steps/45 min PE lesson), or team game lessons (3668 ± 1133 vs. mixed activity lessons; 2759 ± 1183 steps/90 min PE lesson) [38,39]. In game-, fitness-, and skill-oriented lessons in secondary schools, Culpepper and Killion [40] observed on average 2454 ± 716 steps per PE lesson. However, comparisons of reported step counts in PE lessons are problematic due to the wearables used. For example, according to a partial research study carried out in Czech schools using ActiTrainer accelerometers, the average step count achieved by boys was only 1604 ± 784 and by girls 1308 ± 624 steps per 45 min PE lesson [11].

Boys are more physically active in PE lessons than girls, as confirmed by the results of most studies [27,41,42,43]. The main reasons for these gender differences in PE lessons lie in the different content (boys’ content is more fitness-oriented), the fact that fitness lessons are preferred by boys, and the insufficient use of vigorous dance activities by girls, as well as the methods of using team sports (especially soccer among boys and volleyball among girls). We are very critical of the way of teaching and playing volleyball in girls’ PE lessons. Unfortunately, this is based just on observations of PE lessons in schools. This is highly relevant, especially because volleyball is the most popular team game among Czech as well as Polish girls [44,45]. In both countries, much more frequent activities are individual ball games played one-on-one or two-on-two, games on smaller pitches, etc. The promotion of new ways of using volleyball in PE lessons can increase the level of PA in PE lessons and support more efficient use of volleyball in leisure time. The insufficient level of vigorous PA in PE lessons was emphasized by McKenzie et al. [43], who observed only 3.7 min, which represents 14% of the duration of the lesson, and by Smith, Monnat, and Lounsbery [21], who reported 17% of the PE lesson.

### 4.2. Achievement of PE Lesson Recommendations

Hills, Dentel, and Lubans [19] point out that PE lessons do not sufficiently fulfill their educational, motivational, emotional, and motor functions. According to the results of the present study, the number of steps per PE lesson accounts for 21.7% in boys and for 16.8% in girls of the recommended 11,000 steps/day. A study by Smith, Lounsbery, and McKenzie [27] suggests that PE lessons contribute by approximately 25% to the recommended amount of daily PA, but the duration of PE lessons was 27.7 min, i.e., only 65% of the planned duration. Even in our research, especially in girls, it was confirmed that PE lessons do not sufficiently support daily PA.

Concerning the recommendation of 2000 steps per PE lesson, an alarming fact is that almost two thirds of girls do not achieve this minimum number of steps. This finding need not be an indicator of lower PE lesson quality. However, we should think about whether a high-quality PE content, including, for example, more static health-oriented PA, may significantly influence the average step count in PE lessons. Regarding the trends of sedentary behavior in adolescents, the non-achievement of the recommendation of 2000 steps per PE lesson should not be justified by any well-intentioned educational objective in PE [5,17,46].

The recommendation of 2000 steps per PE lesson should be achieved in all locomotion-based PE lessons [11,14]. However, the achievement of this minimum recommendation is not the objective, but merely a reminder of the significance of PA in every PE lesson. In many types of fitness-oriented PE lessons, the achievement of this recommendation is underestimated.

Similarly to the recommendation of 11,000 steps in daily PA, we believe in the importance of the recommended minimum step count in PE lessons jointly for boys and girls. Similar conclusions were also formulated by Silva et al. [47], who also believed that this recommendation should not be distinguished by gender. The main argument for the joint recommendation for boys and girls is the need for a greater focus of girls’ PA in PE lessons on the skeletal system and PA that supports osteoporosis prevention [48]. Another argument is that in weekly PA monitoring, girls’ responses to wearables tend to be more positive compared to boys and that girls probably have more knowledge of physical activity and fitness than boys [49,50].

A well-presented recommendation of 2000 steps per PE lesson in schools in conjunction with other recommendations for specific segments of the school day [11] and adequate use of wearables may provide a more detailed specification of physical activity programs and may support healthy lifestyles in adolescents. Another positive aspect should be the transfer of this approach of PA in PE lessons to other organized forms of PA as part of extracurricular school programs and organized PA leisure activities. PE lessons are a suitable environment to clarify the positives and negatives of wearables and other everyday technologies [51]. This knowledge and experience concerning wearables is also essential for the health and physical literacy of adolescents [52,53,54]. Moreover, PE lessons in all their forms cannot waive their responsibility for physical activity promotion and support [19,20,21], particularly during the current pandemic.

### 4.3. Suggestions for Changes in PE Lessons in the Context of Future PE Development

In the difficult time after the pandemic, the long-standing weaknesses of PE will be more pronounced. Kirk’s [55] criticism of the levels and reforms of PE is highlighted not only in terms of theory, but also from a practical perspective. The existing interaction between the main actors of the educational process—teacher, students, curriculum, and environment—is undergoing significant changes that are ahead of the measures in school practice and especially in PE theory. As a result, it has become extremely difficult to set PE priorities for the current era. In addition, the socioeconomic, demographic, political, educational, historical, ethnic, and other specific regional differences are deepening. Despite this fact, it remains important to define the key priorities for the future development of PE:Cancel exemptions from participation in PE lessons. “Participation in school also means participation in PE lessons or online PE.”Increase the amount and intensity of girls’ PA in PE lessons, particularly through popular PA (dance, dance aerobic, other aerobic activities, etc.).Focus PE lessons on increasing step counts, but also the number of jumps, hops, or dynamic changes in position, which are essential for adolescent girls in the context of osteoporosis prevention and health literacy promotion.Strive to achieve the PA recommendation for PE lessons, i.e., the maximum use of lesson time for the PA preferred by adolescents.In most PE lessons, include at least shorter periods aimed at locomotion activities, including steps, hops, and position changes.In choosing the content of PE lessons, focus on the use and promotion of preferred individual PA that can be performed under movement restrictions.Increase the use of wearables in PE lessons.Introduce online PE as part of distance learning and try to achieve the recommendation of 500 steps/h/lesson and the following recess.In addition to traditional PE lessons, gradually introduce a hybrid system in the curriculum, i.e., a combination of traditional and online PE.Enable adolescents/students to use selective forms of individual fitness testing (in Central European countries, for example, by means of the Indares web-based application).Increase the number of PE lessons in outdoor sports facilities and, where applicable, in a natural environment.Restore traditional forms of body hardening in PE lessons.Encourage students to develop individual PA compensation programs in cooperation with other classmates using wearables and popular information technology.

Future research should focus on clarifying the transfer of the experience of wearables in PE lessons into PA monitoring and achievement of PA recommendations in changed educational conditions. Focus should also be on the benefits of interventions to increase PA in the context of distance learning.

### 4.4. Strengths and Limitations of This Study

The main strength of this study is the measurement of steps per PE lesson in the most real and habitual conditions in the Czech Republic and Poland. The authors assume that the results are very likely to reflect the real conditions in PE lessons in Czech and Polish secondary schools and with a high degree of probability also in Central and Eastern Europe. However, the implementation of the research in habitual school environments limited the monitoring of equally important aspects of successful PE, including educational, social, psychological, and other aspects; as well as the impossibility of objective determination of somatometric characteristics of participants. Another limitation is the selection of secondary schools based on long-standing cooperation with the universities and the location of Polish schools only in the Silesian region. The last limitation concerns the number of participants in teaching practice and school practice.

## 5. Conclusions

This study did not identify any significant differences in step counts per PE lesson in adolescents in teaching practice and school practice. Czech and Polish boys were more physically active in PE lessons than girls. With increasing age, a decrease in step counts was observed only in Czech girls. Weight differences did not have a significant effect on step counts in PE lessons in boys or girls. The use of wearables to monitor step counts in PE lessons and for a simplified estimate of the amount of PA has a solid response value in the natural school environment and has the potential to become a strong stimulus to increase health and physical literacy. The measured data concerning step counts in PE lessons represent an important starting point for the selection of PA promotion strategies and for the future development of PE, as well as for the development and verification of new forms of online PE.

## Figures and Tables

**Figure 1 ijerph-18-01177-f001:**
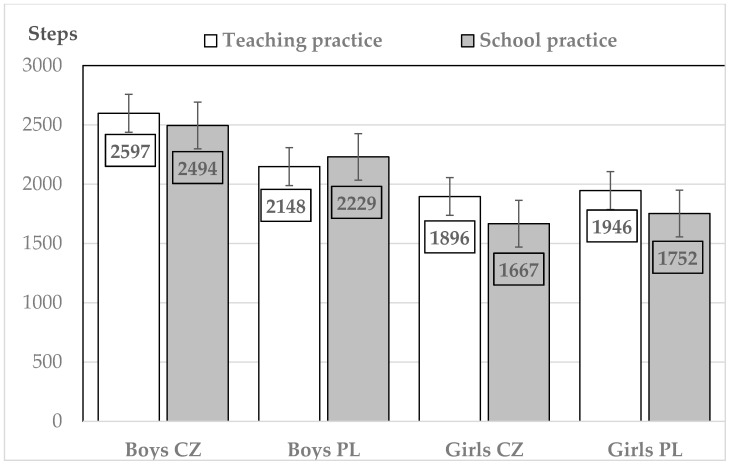
Comparison of step counts per PE lesson (45 min) among Czech (CZ) and Polish (PL) boys and girls during students’ teaching practice and during weekly monitoring of physical activity.

**Figure 2 ijerph-18-01177-f002:**
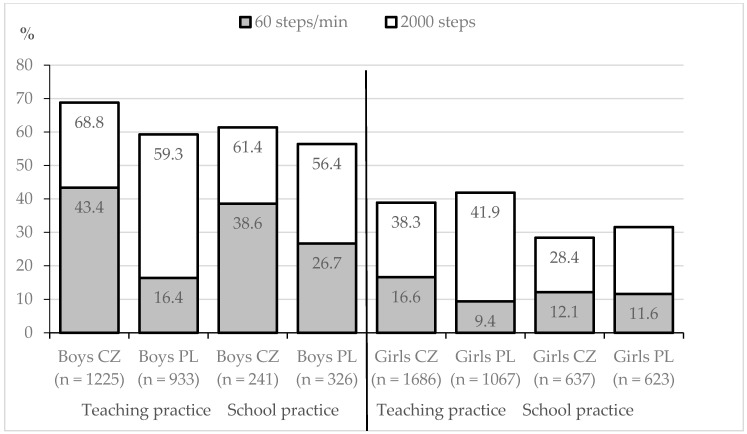
Achievement of the recommendation of 2000 steps and 60 steps/min in PE lesson (45 min) in teaching practice and school practice.

**Table 1 ijerph-18-01177-t001:** Sample characteristics.

Characteristics	*n*	Age (Years) M (SD)	Weight (kg) M (SD)	Height (cm) M (SD)	BMI (kg·m^−2^) M (SD)
Teaching practice					
Boys CZ	1225	16.39 (1.65)	67.98 (12.54)	176.28 (10.08)	21.76 (2.95)
Boys PL	933	16.16 (1.02)	68.89 (12.23)	176.95 (8.21)	21.90 (3.06)
Girls CZ	1686	16.24 (1.50)	55.92 (8.39)	165.93 (6.84)	20.28 (2.60)
Girls PL	1067	16.08 (0.93)	56.15 (8.53)	165.77 (6.12)	20.41 (2.74)
School practice					
Boys CZ	241	15.57 (1.73)	65.68 (15.25)	175.29 (10.96)	21.22 (3.96)
Boys PL	326	16.47 (1.03)	67.04 (11.72)	177.27 (7.61)	21.25 (2.94)
Girls CZ	637	15.89 (1.89)	57.96 (9.64)	166.59 (6.57)	20.83 (2.87)
Girls PL	623	16.65 (0.94)	57.03 (8.66)	166.17 (6.61)	20.61 (2.57)

Note: M = mean; SD = standard deviation; BMI = body mass index; PL = Poland; CZ = Czech Republic.

**Table 2 ijerph-18-01177-t002:** Step counts per PE lesson of boys and girls by age.

Gender	Country	Age	*n*	Steps				
				M	SD	*F*	*p*	*η_p_^2^*
Boys	Czech Republic	<17	768	2638	1149	49.11	<0.001	0.051
		≥17	698	2516	976			
	Poland	<17	844	2216	695			
		≥17	415	2074	758			
Girls	Czech Republic	<17	1276	1935	894	18.21 ^a^	<0.001	0.013
		≥17	1047	1709	869			
	Poland	<17	1011	1919	684			
		≥17	679	1808	718			

Note: M = mean; SD = standard deviation; *F* = ANOVA; *p* = statistical significance; *η_p_^2^* = effect size coefficient. ^a^ Statistically significant differences between younger and older Czech girls.

**Table 3 ijerph-18-01177-t003:** Step counts per PE lesson of boys and girls by BMI.

Gender	Country	BMI (kg·m^−2^)	*n*	Steps				
				M	SD	*F*	*p*	*η_p_* ^2^
Boys	Czech Republic	<25	1285	2603	1076	47.39	<0.001	0.050
		≥25	181	2415	1027			
	Poland	<25	1107	2181	732			
		≥25	152	2082	615			
Girls	Czech Republic	<25	2190	1837	885	1.21	0.305	<0.001
		≥25	133	1774	959			
	Poland	<25	1599	1877	703			
		≥25	91	1827	645			

Note: M = mean; SD = standard deviation; *F* = ANOVA; *p* = statistical significance; *η_p_*^2^ = effect size coefficient.

## Data Availability

The data presented in this study are available on request from the corresponding author.

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
