# Peer review of "Physical Activity Recommendations in the Context of New Calls for Change in Physical Education"

_ijerph, 2021, doi:10.3390/ijerph18031177_

Round 1

Reviewer 1 Report

This study focuses on an important issue. Increasing adolescents’ PA is priority, especially during the pandemic adolescents have attached themselves to electronic devises too much. Hence, possible alternatives to increase adolescents` PA should be investigated.

This paper is well-written paper. I have very minor points to improve the paper. I recommend this paper to be accepted for publication.

Introduction

Please also state your research questions.

Discussion

Please remind readers what the purpose(s) of the study was/were.

Author Response

Comments and Suggestions for Authors

This study focuses on an important issue. Increasing adolescents’ PA is priority, especially during the pandemic adolescents have attached themselves to electronic devises too much. Hence, possible alternatives to increase adolescents` PA should be investigated.

This paper is well-written paper. I have very minor points to improve the paper. I recommend this paper to be accepted for publication.

Introduction

Please also state your research questions.

Discussion

Please remind readers what the purpose(s) of the study was/were.

Response:

Dear reviewer, we would like to thank you for your feedback and valuable recommendations. We revised the title and research question as well as added reminder of the purpose of the study into discussion section.

Reviewer 2 Report

The authors pretend to analyze the contribution of PE to PA in adolescents during and after pandemic.

Being these the objectives, I would recommend altering the title including “recommendations” for PE.

How is it possible that PE will contribute to PA of adolescents during the pandemic when the study was performed during 2015 – 2017? Shouldn’t the study have been conducted during the lockdowns and/or restrictions? Why are the steps assessed years before pandemic and try to extract conclusions?

The objectives are not quite clear to my opinion. Please reformulate.

Abstract:        

L16. “Therefore”? After the introductory sentence this makes no sense. Add age of participants please.

L24-26: not clear. The use of wearables to achieve higher health? Please

consider reading: Finkelstein EA, Haaland BA, Bilger M, Sahasranaman A, Sloan RA, Nang EEK, Evenson KR. Effectiveness of activity trackers with and without incentives to increase physical activity (TRIPPA): a randomised controlled trial. Lancet Diabetes Endocrinol. 2016 Dec;4(12):983-995. doi: 10.1016/S2213-8587(16)30284-4.

Introduction:  

L30-38: Redundant use of “increase” and initial ideas do not come through clearly.

L43-45: Not clear.

L45-50: Why do you include this here? Does it add something relevant to knowledge?

L102-105: Objectives not clear. Please rewrite.

Design and participants:

L108: see my introductory comment on 2015 – 2017.

L118: redundant use of “the sample included”

L122: it is not clear whether PE-curriculum and teaching practices are similar or different in both countries.

L152: Different contents of PE classes lead to different step amount during these classes. If the recording of step counts in PE lessons represented only “one” segment of an active or not-active school-day, then the title and objectives of this study should be modified.

Results:        

L216 – 220: Could you please provide a plausible explanation to the finding that no statistically significant differences were observed in Czech or Polish boys.

L219: Is this sentence correct: “… girls reported, on average more steps per PE lesson than girls …” ?

Discussion:    

L289: Please could you justify why you include the following sentence “We are very critical of the way of teaching and playing volleyball in girls’ PE lessons. Unfortunately, …..” and please explain why you consider this to be important for your study.

L299: “It seems” …? The study only assessed PA in one PE lesson and PA outside of school. What happened to the emotional and motivational functions of PE for PA? Not clear here.

Author Response

Comments and Suggestions for Authors

The authors pretend to analyze the contribution of PE to PA in adolescents during and after pandemic.

Response:

Dear reviewer, we would like to thank you for your feedback and valuable recommendations. We revised the title and research question as well as addressed the purpose of the study in relation to research results. We probably didn't choose the appropriate approach. We were mainly focused on highlighting the main deficiencies in PE, that will be in a new situation even more pronounced.

Being these the objectives, I would recommend altering the title including “recommendations” for PE.

Response:

We revised the title:

Physical Activity Recommendations in the Context of New Calls for Change in Physical Education

How is it possible that PE will contribute to PA of adolescents during the pandemic when the study was performed during 2015 – 2017? Shouldn’t the study have been conducted during the lockdowns and/or restrictions? Why are the steps assessed years before pandemic and try to extract conclusions?

Response:

We revised the concept and objective of the study. Unfortunately, the development of PE in recent years, at least in Central and Eastern Europe, does not sufficiently respect significant changes in the lives of adolescents and society. If PE reforms did not respect the state and the trend before the pandemic, the situation will deteriorate further. It is therefore necessary to highlight simple indicators that can be explained in practice. PE is irreplaceable in explaining the positive and negative tasks of steps count monitoring. Therefore, we tried to establish this as one of the starting points for promoting changes in PE. In addition, similar monitoring of the number of steps in PE lessons to the extent presented will not be feasible in schools in the coming years.

The objectives are not quite clear to my opinion. Please reformulate.

Response:

We revised the objective:

The objective of this study was, to identify the real average step counts in PE lessons among Czech and Polish boys and girls in natural educational environment and propose recommendations for improving PE.

Abstract:        

L16. “Therefore”? After the introductory sentence this makes no sense. Add age of participants please.

Response:

We revised this.

L24-26: not clear. The use of wearables to achieve higher health? Please consider reading: Finkelstein EA, Haaland BA, Bilger M, Sahasranaman A, Sloan RA, Nang EEK, Evenson KR. Effectiveness of activity trackers with and without incentives to increase physical activity (TRIPPA): a randomised controlled trial. Lancet Diabetes Endocrinol. 2016 Dec;4(12):983-995. doi: 10.1016/S2213-8587(16)30284-4.

Response:

We revised this part and used “improve awareness of PA and improve physical literacy” instead. We agree that there is no clear evidence of the benefit of wearables in improving the health of adolescents, but the experience from our (references used in the publication) and other research on the use of wearables bring more positives than negatives.

Introduction:  

L30-38: Redundant use of “increase” and initial ideas do not come through clearly.

L43-45: Not clear.

L45-50: Why do you include this here? Does it add something relevant to knowledge?

Response:

We revised this part and excluded some redundant information. We also tried to clarify the questions and objective of this study.

L102-105: Objectives not clear. Please rewrite.

Response:

We revised this as:

The objective of this study was to identify the real average step counts in Czech and Polish adolescents during PE lessons, in natural educational environment, and propose recommendations for improving PE programs.

Design and participants:

L108: see my introductory comment on 2015 – 2017.

L118: redundant use of “the sample included”

Response:

We adjusted these.

L122: it is not clear whether PE-curriculum and teaching practices are similar or different in both countries.

Response:

We addressed this to make it clear to reader.

“The major difference lies in the PE–curriculum where in Poland the education system includes three PE lessons per week, while the Czech Republic only two PE lessons per week.”

L152: Different contents of PE classes lead to different step amount during these classes. If the recording of step counts in PE lessons represented only “one” segment of an active or not-active school-day, then the title and objectives of this study should be modified.

Response:

We agree and addressed this already

Results:        

L216 – 220: Could you please provide a plausible explanation to the finding that no statistically significant differences were observed in Czech or Polish boys.

Response:

We did not make this as a main finding. It is just interesting, that number of steps per PE class did not vary if the class was led by school teacher with long term experience compare to student teacher leading. In contrary in girls these differences were significant in favor to student teachers!

L219: Is this sentence correct: “… girls reported, on average more steps per PE lesson than girls …” ?

Response:

Yes, this is correct, we just added clarification of nationality.

Discussion:    

L289: Please could you justify why you include the following sentence “We are very critical of the way of teaching and playing volleyball in girls’ PE lessons. Unfortunately, …..” and please explain why you consider this to be important for your study.

Response:

We addressed Teaching volleyball in schools tends to respects the sports rules rather than adapt this popular game to school environment. Due to the high number of volleyball lessons in schools, the requirements to increase their effectiveness in terms of the level of PA are very crucial. We added “The promotion of new ways of using volleyball in PE lessons can increase the level of PA in PE lessons and support more efficient use of volleyball in leisure.”

L299: “It seems” …? The study only assessed PA in one PE lesson and PA outside of school. What happened to the emotional and motivational functions of PE for PA? Not clear here.

Response:

We modified “Hills, Dentel, and Lubans [21] point out that PE lessons do not sufficiently fulfill their educational, motivational, emotional and motor functions.” Furthermore we added “Even in our research, especially in girls, it was confirmed that PE lessons do not sufficiently support daily PA.”

Reviewer 3 Report

This is an interesting study involving analysis of physical activity in adolescents. The study is well written and have sound methods, with a nice sample size.

Title

The study was performed between 2015 and 2017, so it has nothing to do with the COVID-19 Pandemic.

Introduction

The study has nothing to with the COVID Pandemic. I suggest improving the rationale in this context and remove any reference to the Coronovirus outbreak.

Methods

Step count might underestimate physic activity levels when the classes involve activities that do not have displacement, like calisthenics. For example, swimming (line 175) might be a popular physical activity mode, did you control for this?

Discussion and conclusion.

I suggest removing the topic 4.3. Maybe you can present that in another article. I also suggest to remove reference to COVID-19 in the conclusions.

Author Response

Comments and Suggestions for Authors

This is an interesting study involving analysis of physical activity in adolescents. The study is well written and have sound methods, with a nice sample size.

Response:

Dear reviewer, we would like to thank you for your feedback and valuable recommendations. We revised the title and research question as well as addressed the purpose of the study in relation to research results.

Title

The study was performed between 2015 and 2017, so it has nothing to do with the COVID-19 Pandemic.

Response:

We addressed this important note. First we wanted to point out that when challenged to serious problems in PE during and after the pandemic, we should respect and learn from the state and trend in PE before the pandemic. Especially when school PE in recent years was not responding enough the acceleration of changes in the lifestyle of adolescents.

Introduction

The study has nothing to with the COVID Pandemic. I suggest improving the rationale in this context and remove any reference to the Coronovirus outbreak.

Response:

We addressed this throughout the text.

Methods

Step count might underestimate physic activity levels when the classes involve activities that do not have displacement, like calisthenics.

Response:

Thank you for this important point we did not mentioned.  Achieving a certain number of steps in PE lessons is a simple goal and must not be the main goal. We try to draw attention to the need for locomotor activities (there is a good indicator of the number of steps, hops and changes in position), but not at the expense of more physically static or medically and mentally oriented activities.

For example, swimming (line 175) might be a popular physical activity mode, did you control for this?

Response:

We only included PE lessons that allowed using wearables and also keep the process most habitual as possible. We added “To compare teaching practice within countries, only step counts in PE lessons were used from the weekly PA monitoring” Swimming lessons were not on the program at the time of the research.

Discussion and conclusion.

I suggest removing the topic 4.3.

Response:

We did not want to be satisfied only with stating the situation in PE lessons. We modified this part as future vision in PE lessons and challenges that are coming based on educational, cultural and other changes in the society. We consider these suggestions to be relevant for school practice.

Maybe you can present that in another article. I also suggest to remove reference to COVID-19 in the conclusions.

Response:

We removed COVID-19 throughout the text.

Reviewer 4 Report

The authors raise an important problem related to the current state of physical activity of school youth and the role of physical education in the context of international research. In my opinion, they also indirectly prepare teachers to monitor the intensity level of physical education classes.

I would like to ask about possible limitations of your research:

  1. Lines 35-38 – „There has been a long-term downward trend in physical activity (PA) of adolescents [1,2], as well as an increase in sedentary behavior [3–5], an increase in overweight and obesity [6,7], increasing insufficient cardiorespiratory fitness [8], and negative trends in mental health [9], including other undesirable effects on adolescents’ health”

This is of course the most appropriate point. Therefore, my question concerns the selection of the group for the research - why in your study the number of overweight people (BMI≥25) is only 8.3% of all respondents (8.5% girls, 8.1% boys). You also do not provide data on the number of people with obesity BMI≥30 (were there no such people, people with obesity do not participate in physical education classes, are exempt from classes, did not consent to the study?) BMI≥25 in people practicing sports may not be a sign of being overweight due to body fat, but a higher level of muscle mass (haven't you considered adding waist and hip measurements, determining other metrics such as WHR, WHtR?). I believe that underweight people should also be mentioned to better illustrate the study group.

  1. The pedometer is a valid instrument when estimating physical activity levels, but caution is urged when interpreting movements other than walking. Classes, based on jumping exercises, can be very intense, which is confirmed by heart rate monitors set up for students during PE lessons. Therefore, have you not considered asking students for a subjective assessment of the intensity of the activities (Borg scale)?
  2. What were the technical problems with putting on pedometers, starting classes on time, and whether it was taken into account in the average duration of the lesson (45min).
  3. Physical education at school is not attractive to girls as it relies heavily on team games. Have you researched the intensity of different types of physical education lessons, e.g. based on the H-RF (Health-Reladed Fitness) concepts?
  4. Research shows that a significant number of physical education teachers have difficulty controlling the intensity of their lessons. Therefore, it seems that also a practical conclusion (recommendation) of this article is the need to train teachers in the selection of intensity and the use of modern forms of its monitoring.

Author Response

Comments and Suggestions for Authors

The authors raise an important problem related to the current state of physical activity of school youth and the role of physical education in the context of international research. In my opinion, they also indirectly prepare teachers to monitor the intensity level of physical education classes.

Response:

Dear reviewer, we would like to thank you for your feedback and valuable recommendations. We revised the paper in relation to adequately address the purpose of the study in relation to research results. We were limited in this particular study to subjective somatometric data, so not so many details as you mentioned could be presented here. Other types of studies including objective measures of somatic data are being done within our research group, however this study solves different task.

I would like to ask about possible limitations of your research:

  1. Lines 35-38 – „There has been a long-term downward trend in physical activity (PA) of adolescents [1,2], as well as an increase in sedentary behavior [3–5], an increase in overweight and obesity [6,7], increasing insufficient cardiorespiratory fitness [8], and negative trends in mental health [9], including other undesirable effects on adolescents’ health”

This is of course the most appropriate point. Therefore, my question concerns the selection of the group for the research - why in your study the number of overweight people (BMI≥25) is only 8.3% of all respondents (8.5% girls, 8.1% boys). You also do not provide data on the number of people with obesity BMI≥30 (were there no such people, people with obesity do not participate in physical education classes, are exempt from classes, did not consent to the study?) BMI≥25 in people practicing sports may not be a sign of being overweight due to body fat, but a higher level of muscle mass (haven't you considered adding waist and hip measurements, determining other metrics such as WHR, WHtR?). I believe that underweight people should also be mentioned to better illustrate the study group.

Response:

Dear reviewer, this is the crucial question and is correct. However boys and girls who are overweight or obese often do not participate in PE lessons. Our data on overweight and obesity (CZ boys 12.4%, PL boys 12.1%, CZ girls 5.7% and PL girls 5.4%) are lower than partial results, eg from HBSC in 15-year-old boys 20.8% and girls 10.9% (HamĹ™ík , Z. et al., Trends in Overweight and Obesity in Czech Schoolchildren from 1998 to 2014. Cent Eur J Public Health, 2017, 25 (1): S10-S14). Unfortunately, we do not have exact data on the numbers of boys and girls who were completely or partially exempted from PE for individual reasons at individual schools. In the Czech Republic and Poland, parents can excuse non-participation in PE-lessons (Physical Education and Sport at School in Europe, 2013), which is the biggest problem. Based on official pupil lists, the number of pupils who did not participate in PE lessons in the research ranged from 10% to 20%. Monitoring of step counts in PE lessons in teaching practice was completed by all boys and girls who were at school on the given day and participated in PE lessons. In weekly PA monitoring, on average only 1 to 3 pupils in one class refused to participate in the study (n = 25). We added in the methodology and in the limits: ”A total of 12.2% of boys and 5.6% of girls were overweight or obese (14.1% of boys and 22.7% of girls were underweight).”

  1. The pedometer is a valid instrument when estimating physical activity levels, but caution is urged when interpreting movements other than walking. Classes, based on jumping exercises, can be very intense, which is confirmed by heart rate monitors set up for students during PE lessons. Therefore, have you not considered asking students for a subjective assessment of the intensity of the activities (Borg scale)?

Response:

Dear reviewer this was not possible within the aims of this research.

  1. What were the technical problems with putting on pedometers, starting classes on time, and whether it was taken into account in the average duration of the lesson (45min).

Response:

There were no problems starting and ending PE lessons. PE lessons were recalculated to 45 minutes.

  1. Physical education at school is not attractive to girls as it relies heavily on team games. Have you researched the intensity of different types of physical education lessons, e.g. based on the H-RF (Health-Reladed Fitness) concepts?

Response:

This was not possible in an effort not to interfere with PE lessons, however HR monitoring is extensively used in other research we realize.

  1. Research shows that a significant number of physical education teachers have difficulty controlling the intensity of their lessons. Therefore, it seems that also a practical conclusion (recommendation) of this article is the need to train teachers in the selection of intensity and the use of modern forms of its monitoring.

Response:

This comment has a crucial impact. The results of the research we always use in the professional training of PE teachers at university workplaces in the Czech Republic and Poland.

Round 2

Reviewer 2 Report

Thank you attending all the proposed changes.